# The Impact of the COVID-19 Pandemic on the Number of Cancer Patients and Radiotherapy Procedures in the Warmia and Masuria Voivodeship

Marcin Kurowicki [1], Karolina Osowiecka [2,*], Barbara Szostakiewicz [1], Monika Rucińska [3] and Sergiusz Nawrocki [3,4]

1   NU-MED Radiotherapy Center in Elblag, Królewiecka 146, 82-300 Elblag, Poland
2   Department of Psychology and Sociology of Health and Public Health, School of Public Health, University of Warmia and Mazury in Olsztyn, Warszawska 30, 10-082 Olsztyn, Poland
3   Department of Oncology, Collegium Medicum University of Warmia and Mazury in Olsztyn, Wojska Polskiego 37, 10-228 Olsztyn, Poland
4   Department of Radiotherapy, Hospital of the Ministry of Internal Affairs with Warmia and Mazury Oncology Center in Olsztyn, Wojska Polskiego 37, 10-228 Olsztyn, Poland
*   Correspondence: karolina.osowiecka@uwm.edu.pl

**Abstract:** (1) Background: It was suspected that the COVID-19 pandemic would negatively affect health care, including cancer treatment. The aim of the study was to assess the impact of the COVID-19 pandemic on the number of radiotherapy procedures and patients treated with radical and palliative radiotherapy in Poland. (2) Methods: The study was carried out in Warmia and Masuria voivodeship. The number of procedures and treated patients one year before and in the first year of the COVID-19 pandemic were compared. (3) Results: In the first year of the COVID-19 pandemic, the number of radiotherapy procedures and cancer patients treated with radiotherapy in Warmia and Masuria voivodeship in Poland was stable compared to the period before the pandemic. The COVID-19 pandemic has not affected the ratio of palliative to radical procedures. The percentage of ambulatory and hostel procedures significantly increased with the reduction of inpatient care in the first year of the COVID-19 pandemic. (4) Conclusion: No significant decrease in patients treated with radiotherapy during the first year of the pandemic in Warmia and Masuria voivodeship in Poland could indicate the rapid adaptation of radiotherapy centers to the pandemic situation. Future studies should be carried out to monitor the situation because the adverse effects of the pandemic may be delayed.

**Keywords:** cancer; radiotherapy; COVID-19; healthcare; pandemic

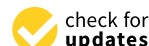



## 1. Introduction

The pandemic broke out suddenly and came as a surprise to the whole world. The first unusual cases of pneumonitis were noted in Wuhan, China, in December 2019. A hitherto unknown version of the coronavirus was isolated from the material collected from patients: SARS-CoV-2 [1]. The virus began to spread rapidly worldwide. The first cases of a new illness called COVID-19 in Europe were observed in France in January 2020 [2]. In Poland, the first case of COVID-19 was reported on 4 March 2020, in Zielona Góra [3]. Most countries introduced various restrictions to reduce the number of infected patients. Gradually, isolation, quarantine, limits on people staying in public places, and transport were incorporated. Mass events were cancelled, and hand disinfection and mask-wearing were ordered. On 20 March 2020, an epidemic state in Poland was declared [4]. An order to stay home was in force from 25 March to 11 April.

The pandemic situation was a challenge for the healthcare system. There were no established procedures and organizations for dealing with a pandemic. The medical system

faced many problems related to the increasing number of infected patients. Access to health care for patients diagnosed with different diseases, including cancer patients, has become limited. Diagnostic departments such as Internal Medicine, Surgical and Lung Diseases Departments, and even entire hospitals were transformed into units dealing with only COVID-19 patients. The pandemic also caused staff shortages. Some medical staff were infected; some were quarantined, and some were redirected to fight the epidemic. Therefore, the Ministry of Health allowed telecommunication with patients in primary and specialist health care.

On the other hand, the number of cancer patients did not decrease. According to the scientific medical communities' recommendations, the priority was to treat only emergency illness and refrain from treating chronic diseases in the first days of the pandemic. However, the waiting time for cancer diagnosis and treatment is adverse. The longer waiting time for cancer diagnosis and treatment influences higher mortality and lower survival and correlates with more advanced stages of disease among breast cancer patients [5,6]. Jensen et al. [7] showed a negative impact of waiting time on tumor progression in patients with head and neck cancer. Patients diagnosed at more advanced stages have a poorer prognosis and are less likely to receive radical treatment with the intention of a cure. Due to pandemic restrictions, there were concerns that cancer patients were not receiving adequate treatment, or/and they were treated at more advanced clinical stages. In Poland, the number of screening tests decreased. Initially, the National Health Fund decided to suspend or limit diagnostic and prophylactic tests, such as mammography, to reduce the risk of virus transmission [8]. The study conducted by Andrzejczak et al. [9] confirmed that introduced pandemic restrictions had a negative impact on patients' participation in cancer screening. The number of mammography tests in the breast cancer screening program was reduced by over 90%, and cytology tests in the cervical cancer prevention screening program decreased by more than 85% [9].

Therefore, there was an idea to analyze the number of cancer patients and radiotherapy procedures during the first year of the COVID-19 pandemic and to compare it to the time before the pandemic. The study aimed to assess the impact of the COVID-19 pandemic on the number of radiotherapy procedures and the number of cancer patients treated with radical and palliative radiotherapy in the Warmia and Masuria voivodeship.

## 2. Materials and Methods

In Warmia and Masuria voivodeship, cancer patients are treated using radiation therapy in two oncological centers. The study was carried out in these two centers: Hospital of the Ministry of Internal Affairs with Warmia and Mazury Oncology Center in Olsztyn and the NU-MED Radiotherapy Center by the Provincial Integrated Hospital in Elbląg. The data for the analysis were generated from hospitals' databases. The obtained data were anonymized. In the study, radiotherapy procedures from 1 March 2019 to 28 February 2021 was analyzed. Brachytherapy procedures were excluded from the analysis. One patient could have more than one procedure during the analyzed period. Therefore, two separate analyses were conducted in the study: the number of radiotherapy procedures and the number of patients treated with radiation therapy. The differences in the number of procedures and treated patients between the time before the COVID-19 pandemic (1 March 2019–29 February 2020) and the first year of the COVID-19 pandemic (1 March 2020–28 February 2021) were determined. The impact of the COVID-19 pandemic on the number of patients treated with radiation therapy and radiotherapy procedures was estimated due to cancer localization according to ICD-10 classification, the intention of treatment (radical or palliative), type of care (inpatient, outpatient, or hostel), treatment under the rapid oncology pathway (Diagnostic and Treatment of Cancer; DiLO). The data on COVID-19 incidence during the first year of the pandemic were received from Provincial Sanitary and Epidemiological Station in Olsztyn. Approval was received from the centers to conduct the study.

*Statistical Analysis*

Descriptive statistics were used to determine the characteristic. The Chi-square test was used to compare the prevalence of radiotherapy procedures/cancer patients treated with radiotherapy before and during the COVID-19 pandemic due to various factors. A *p*-value < 0.05 was considered to be significant. The analysis was conducted using Statistica (data analysis software), version 13. http://statistica.io (accessed on 1 November 2021) TIBCO Software Inc., Krakow, Poland (2017).

## 3. Results

In the last year before the COVID-19 pandemic (1 March 2019–29 February 2020), 3161 radiotherapy (RT) procedures, including 1793 radical and 1368 palliative, were carried out in Warmia and Masuria voivodeship. During the first year of the COVID-19 pandemic (1 March 2020–28 February 2021), 2902 RT procedures were provided (an 8% decrease from the previous year), including 1690 radical RT and 1212 palliative RT procedures (respectively, 6% and 11% decrease from the previous year).

In Elbląg, only a 2.9% decrease from the previous year in RT procedures was observed during the pandemic, whereas in Olsztyn, 12.2% fewer than the previous year RT procedures were conducted. The difference in RT procedures before and during the pandemic between Olsztyn and Elbląg was insignificant. There were insignificant changes in the percentage of RT procedures due to the intention of RT procedures in both centers (Figure 1A,B).

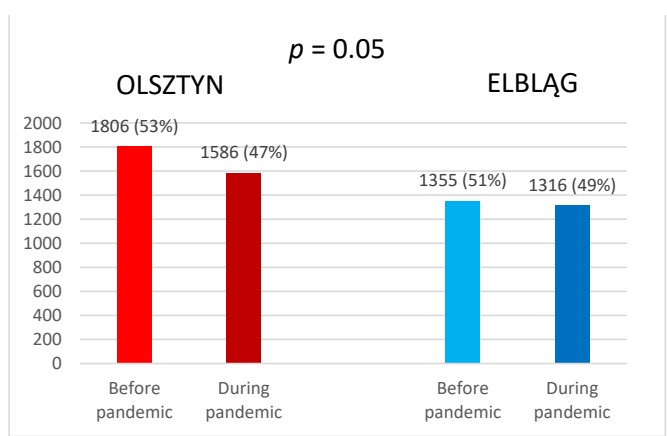

(**A**)

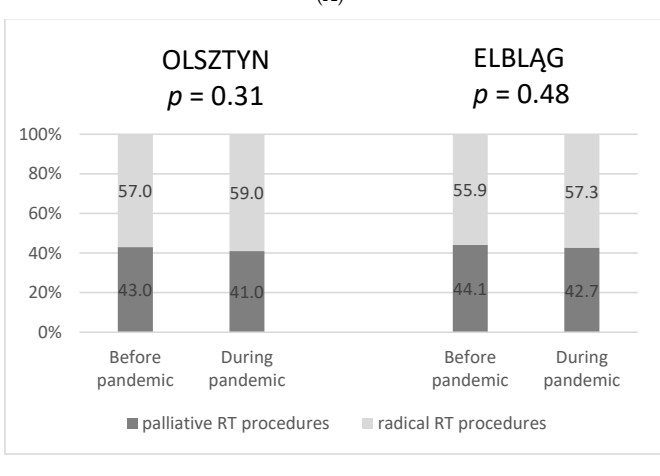

(**B**)

**Figure 1.** Comparison of radiotherapy procedures before and during the pandemic between Olsztyn and Elbląg (**A**). Comparison of percentage of RT procedures due to radical and palliative intention in Olsztyn and Elbląg (**B**).

Two thousand seven hundred fifty-two patients were treated with radiotherapy in the last year before the COVID-19 pandemic, and 2514 patients during the first year of the pandemic in the Warmia and Masuria voivodeship. The number of patients undergoing RT was reduced insignificantly by 3.6% and 12.2% compared to the previous year, respectively, in Elbląg and Olsztyn. There was no significant difference between Olsztyn and Elbląg. The pandemic did not influence the number of cancer patients treated with palliative and radical intention (Figure 2A,B).

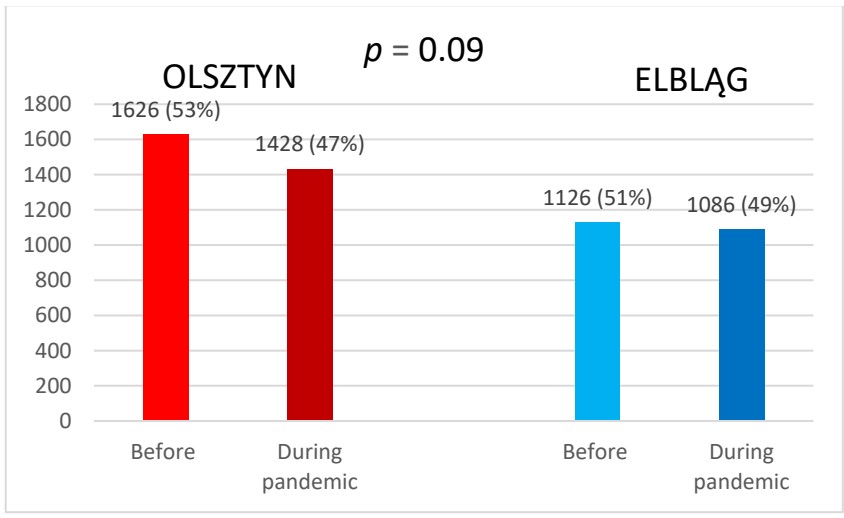

(**A**)

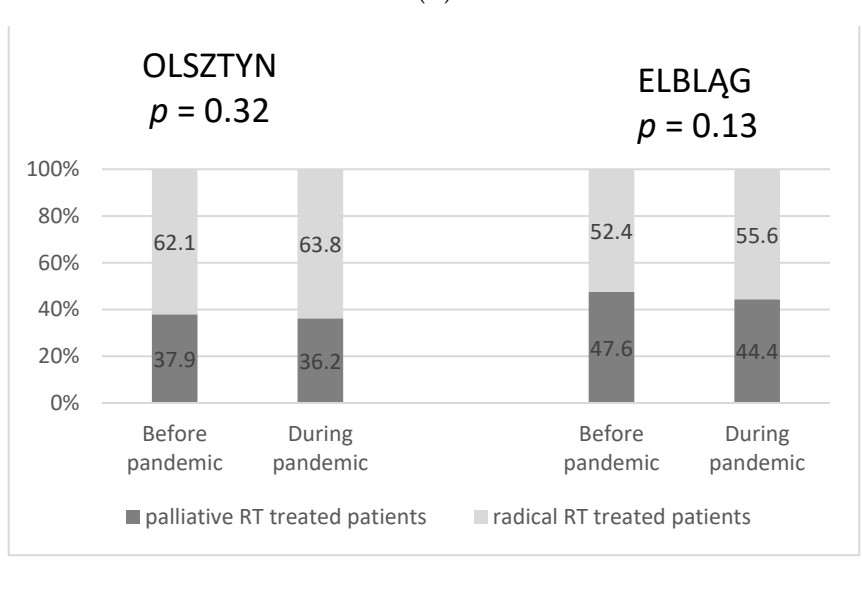

(**B**)

**Figure 2.** Comparison of patients treated with radiotherapy before and during the pandemic between Olsztyn and Elbląg (**A**). Comparison of percentage of patients treated with radical and palliative radiotherapy in Olsztyn and Elbląg (**B**).

In both centers, the percentage of ambulatory and hostel procedures significantly increased with the reduction of inpatient care during the pandemic compared to the previous year (Figure 3).

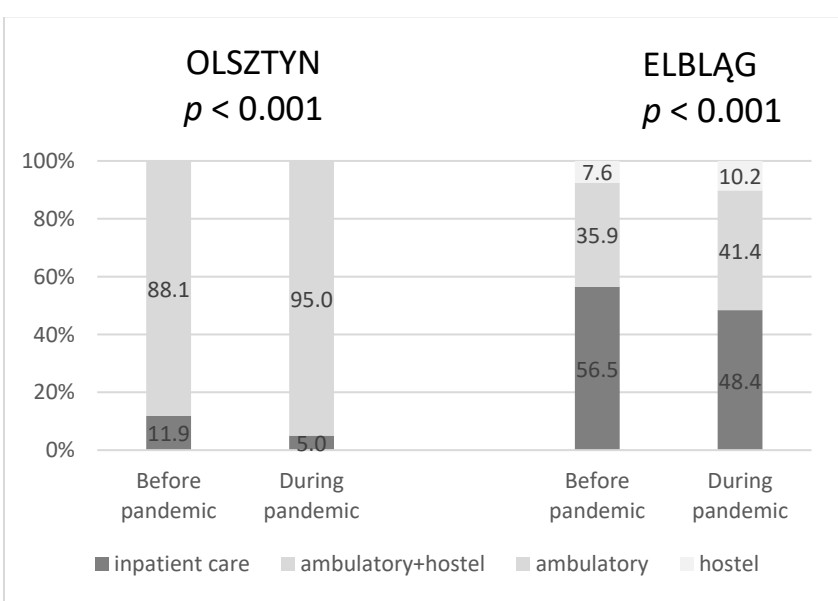

**Figure 3.** The percentage of procedures due to different types of a patient's care before and during the pandemic in Olsztyn and Elbląg.

There was no significant difference in the percentage of patients treated under the rapid oncology pathway (DiLO) vs. without this pathway during the first year of the COVID-19 pandemic compared to the previous year. (Figure 4).

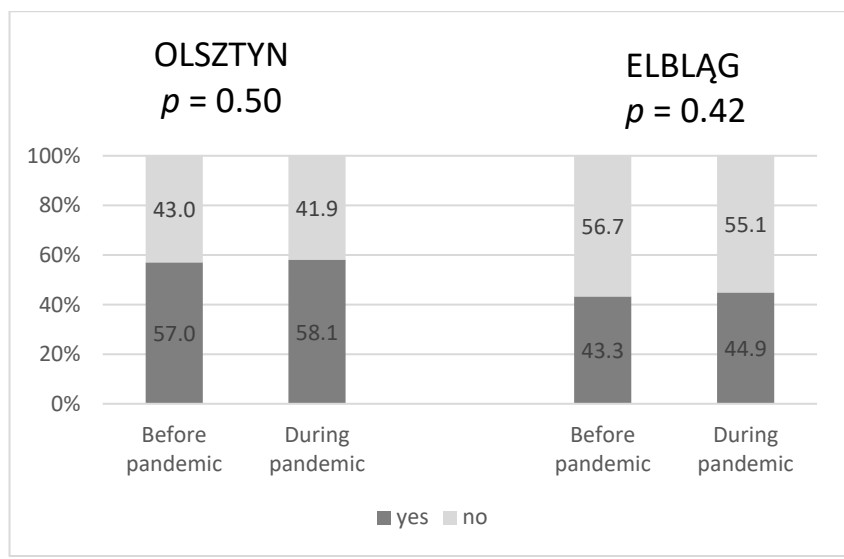

**Figure 4.** The percentage of patients treated with radiotherapy under DiLO before and during the pandemic in Olsztyn and Elbląg.

The distribution of the total number of procedures and patients treated with RT in Olsztyn and Elbląg in 2 years showed significant differences due to cancer localization between the two centers ($p < 0.05$). In Elbląg, more RT procedures were conducted among lung cancer patients, whereas in Olsztyn, among breast cancer patients (Table S1 in Supplementary Material).

There were significant changes in the proportion of radiotherapy procedures/patients treated with radiation therapy in the case of head and neck cancers, breast cancers, lower digestive system cancers, and FPI. In the Warmia and Masuria voivodeship, there was a 33% and 38% decrease in palliative RT procedures and patients with head and neck cancer, respectively. There was a reduction of 26% of palliative RT procedures and 3% of radical RT procedures among breast cancer patients ($p = 0.02$) in the Warmia and Masuria voivodeship.

In Elbląg, there were a significant decrease in palliative (by 39%) and an increase in radical RT procedures (by 27%) for lower digestive system cancer patients. In Elbląg number of palliative RT procedures and patients treated with FPI cancer was significantly lower during the pandemic. In the case of upper digestive system cancers, respiratory system cancers, gynecological cancers, urinary system cancers, prostate cancers, and CNS cancer, no significant differences were reported between radical and palliative RT procedures and patients proportions due to the pandemic (Tables S2–S4 in Supplementary Material, Figures S1 and S2 in Supplementary Material).

In Warmia and Masuria voivodeship, during the first year of the pandemic, the incidence of COVID-19 increased from 56 cases to above 15,000, whereas the RT procedures decreased by about 25%. A decrease in RT procedures was observed in the months of the highest incidence of COVID-19 (November–December) and continued in the following two months. However, this decrease was not proportionally correlated with the increase in COVID-19 incidence (Figure 5).

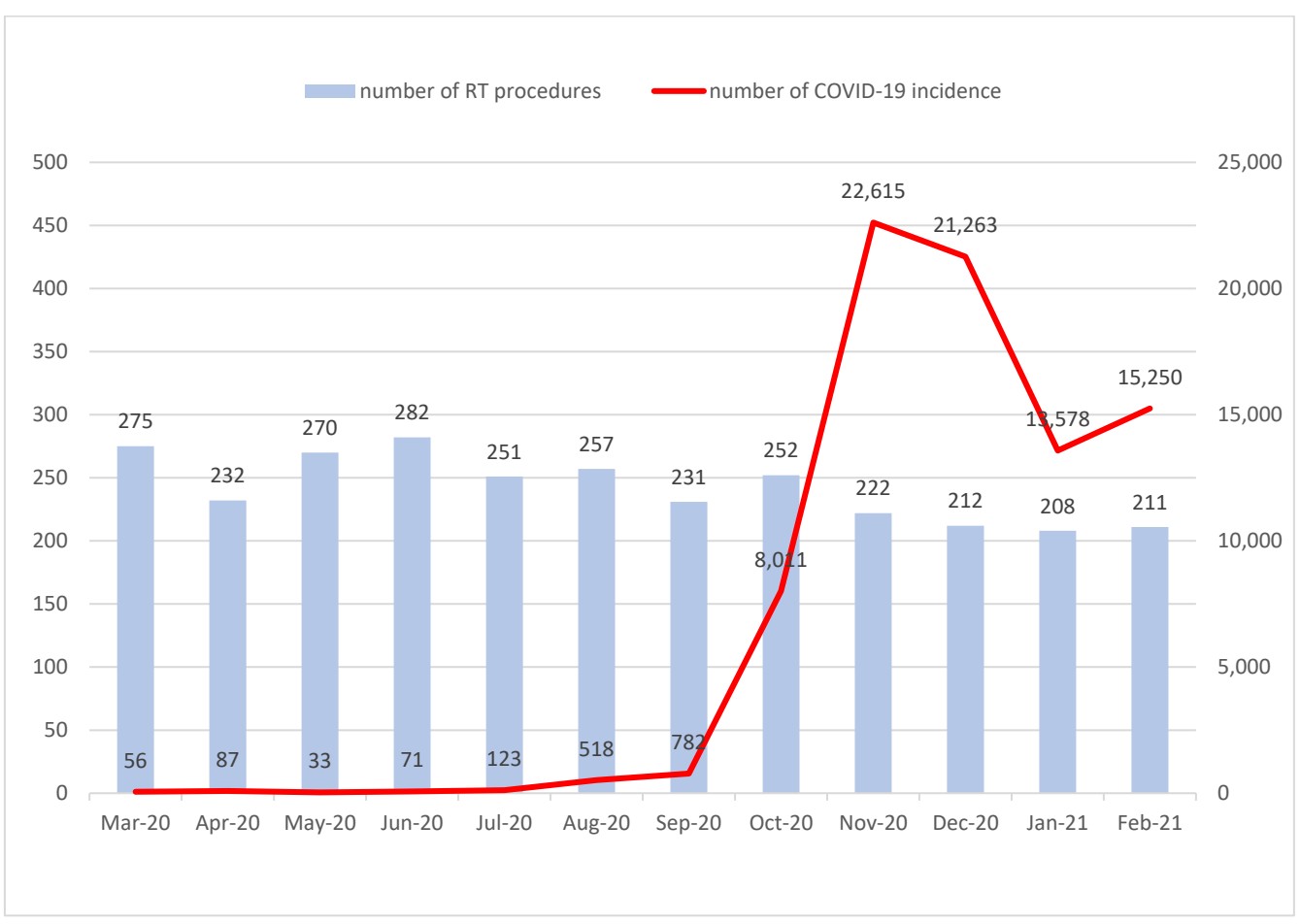

**Figure 5.** The correlation between the number of RT procedures in Warmia and Masuria voivodeship and the COVID-19 incidence during the first year of the pandemic.

## 4. Discussion

The world was adapting to life in a pandemic. Vaccinations against COVID-19 were introduced on 27 December 2020 [10]. From January 2021, limitations were lightened gradually again. Until this publication in Poland, three more waves were observed, with peaks in March/April 2021, November/December 2021, and January/February 2022 and this time with fewer restrictions recommended [11]. At the end of April 2022, the regulations began to be lifted gradually [12]. Due to the second wave of the disease, restrictions were announced again throughout Poland. After the announcement of the epidemic in

Poland in March 2020, each healthcare unit had to develop various systems for coping with the situation. At the beginning of the first wave of COVID-19 cases, there were no consistent recommendations regarding work logistics in hospital departments and diagnostic or therapeutic procedures. Each establishment had to prepare procedures that would allow for the continuity of patient care but at the same time would not expose these patients and working staff to SARS-CoV-2 infection. The pandemic was a challenge for the healthcare system and required an organized effort by all medical staff. Similar sanitary methods to limit the spread of the virus have been introduced in Poland and throughout Europe [13–15]. It would seem that fewer patients underwent oncological treatment or/and they would be treated at more advanced stages. In our study, the number of RT procedures and patients treated with RT decreased during the first year of the pandemic in Warmia and Masuria voivodeship and at each center, but the differences were not significant. The observed reduction of the number of RT procedures and patients treated with radiotherapy in Warmia and Masuria voivodeship as compared to the previous year was 8% and 9%, respectively.

There is a lack of studies on the impact of the COVID-19 pandemic on radiotherapy, especially during its first year. In our study, we noted a decrease in RT procedures in months of the highest incidence of COVID-19 during the first year of the pandemic, but this decrease was not proportionally associated with an increase in COVID-19 cases. Despite severe restrictions introduced at the beginning of the pandemic, the decrease in radiotherapy procedures was low. In the Greater Poland Cancer Center study, a decrease in patients starting radiotherapy by as much as 44% was temporarily observed [15]. We also analyzed data from the "Report on the state of radiotherapy in Poland as of 31 December 2020." published by the national consultant for oncological radiotherapy comparing 2020 to 2019 [16]. In 2020 in Poland, a decrease in the number of patients receiving radiotherapy by 5% was observed as compared to the previous year. The most significant decrease was observed in Podlasie voivodeship (approx. 14%) as well as Lesser Poland voivodeship and Opole voivodeship (approx. 11%). Some voivodeships, despite the difficulties, noted an increase in the number of patients—Lubusz voivodeship (approx. 8%), Lower Silesia, and Kujawy-Pomerania voivodeships (approx. 2%). The decrease in the number of patients treated with radiotherapy in the Warmia and Masuria voivodeship was 7% as compared to the previous year. The situation was similar in other countries. During the first year of the pandemic, there were noted drops in the patients treated with radiotherapy by 6–36% in Europe, 20–32% in North America, and 8% in Latin America [13,17–23].

Due to the observed delays in the diagnosis of neoplastic diseases and the reluctance of patients to undergo diagnostics for fear of being infected with the SARS-CoV-2 virus, we were afraid of an increase in the number of palliative radiotherapies. From March to May 2020, the center in Pisa observed an increase in palliative radiotherapy by about 30% compared to the previous year [18]. In the current study, there was no statistically significant increase in the number of patients treated with radiotherapy with palliative intent. Probably it is too early to detect differences. Many cancer patients in the current study were diagnosed and started their treatment before the pandemic. Maybe differences will be noted in the following years due to the delay and difficulty accessing a doctor and screening tests.

It is worth noting that both centers, despite many difficulties, have not decided to limit the number of patients admitted. No significant differences in the number of RT procedures and patients undergoing treatment (radical and palliative) indicated a good organization at both centers. The number of hospitalizations was limited only to the necessary ones. The significant increase in the number of outpatients may prove an excellent adaptation to the pandemic situation by both centers.

Comparing 2020 to 2019 in Poland, the number of patients involved into DiLO, number of consultations, visits to primary health care centers, first-time specialist clinic visits, and the number of hospitalizations among cancer patients (ICD 10 code starting with C)

dropped off radically [24]. In the current study, there was no significant difference in the number of patients treated under DILO during the first year of COVID-19 pandemic.

A special effort to minimize the risk of viral transmission by patients during oncological treatment in radiotherapy departments was to shorten their treatment time by introducing moderately hypofractionated regimens in radical radiotherapy. This treatment should be of comparable efficacy and toxicity as in the case of classically fractionated regimens. Moreover, in palliative radiotherapy, the shortening of schedules was also presented. During the pandemic, foreign and Polish guidelines recommended appropriate actions appeared gradually [25–30].

In Warmia and Masuria region, the significant difference in the number of patients treated with radical and palliative radiotherapy was observed only in the case of head and neck cancer tumors—a decreased number of patients treated palliatively by 38% and an increased number of patients treated radically by 26% as compared to the previous year. A decrease of 44% of head and neck cancer patients who underwent palliative radiation in Elbląg was reported, whereas an increase in radical patients by 26% compared to the previous year. It could be a reason for the introduction of moderate hypofractionation (50 Gy in 20 fractions, 2.5 Gy per fraction) in patients not eligible for surgery and standard radical RT and, at the same time, in too good condition for standard palliative radiotherapy.

In Elbląg, there was observed an increase in the number of radically treated patients with cancer of the lower digestive system. Similar observations were made in the UK [21]. Despite the limitations in breast cancer screening tests, there were no differences in the number of radical (adjuvant) radiotherapy procedures; however, the number of palliative procedures was significantly reduced in breast cancer patients compared to the previous year.

A decrease in the number of prostate cancer patients treated with radiotherapy and radiotherapy procedures was reported in UK or Lithuania [21,22,31]. There were no such differences in Warmia and Masuria voivodeship compared to the previous year.

## 5. Conclusions

In the first year of the COVID-19 pandemic, the number of cancer patients treated with radiotherapy in the Warmia–Masuria voivodeship was stable. However, during the months of the highest COVID-19 incidence, a low decrease in RT procedures was noted, which was not proportionally correlated with the increase in COVID-19 cases. There was no significant decrease in the total number of cancer patients treated with radiotherapy and an increase in the number of cancer patients treated with radiotherapy with palliative intent compared to the previous year. It could indicate an adaptation to the pandemic situation and good organization of the health care system in the Warmia–Masuria voivodeship. The analysis included only the first year of the pandemic. Future studies should be carried out to monitor the situation because the adverse effects may be delayed.

**Supplementary Materials:** The following supporting information can be downloaded at: https://www.mdpi.com/article/10.3390/curroncol30010077/s1, Table S1: Distribution of RT procedures and patients due to cancer localization. Table S2: The number of radiotherapy procedures and patients treated with radiation therapy in the Warmia-Masuria voivodeship due to different cancer localization. Table S3. The number of radiotherapy procedures and patients treated with radiation therapy in Olsztyn due to different cancer localization. Table S4. The number of radiotherapy procedures and patients treated with radiation therapy in Elbląg due to different cancer localization. Figure S1. Distribution of RT procedures due to cancer localization. Figure S2. Distribution of patients treated with RT due to cancer localization.

**Author Contributions:** Conceptualization, M.K., B.S. and S.N.; Data curation, M.K., K.O. and M.R.; Formal analysis, M.K., K.O. and M.R.; Investigation, M.K., K.O. and B.S.; Methodology, M.K., K.O., B.S. and S.N.; Project administration, M.K. and K.O.; Resources, M.K. and K.O.; Supervision, S.N.; Visualization, K.O. and M.R.; Writing—original draft, M.K., K.O. and M.R.; Writing—review & editing, K.O. and M.R. All authors have read and agreed to the published version of the manuscript.

**Funding:** This research received no external funding.

**Institutional Review Board Statement:** Not applicable.

**Informed Consent Statement:** Informed consent was obtained from all subjects involved in the study.

**Data Availability Statement:** The data presented in this study are available on request from the corresponding author.

**Conflicts of Interest:** The authors declare no conflict of interest.

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
