# Peer review of "The Impact of the COVID-19 Pandemic on the Number of Cancer Patients and Radiotherapy Procedures in the Warmia and Masuria Voivodeship"

_curroncol, doi:10.3390/curroncol30010077_

Round 1

Reviewer 1 Report

On line 153 the term DiLO appears, but there is no sufficient explanation of what it means. Tables 3 and 4 seem to me not very clear and the information contained is not interesting for the reader in this form. Can you transfer them to supplementary materials?

Minor editorial errors, e.g. no space in line 292, inconsistent dots after citing literature.

Author Response

Thank you very much for reviewing our manuscript and for your kind comments.

Ad. 1 DiLO means Diagnostic and Treatment of Cancer – polish name of pathway. In line 94 in Material and Methods section we added explanation.

In line 159 we corrected – “under the rapid oncology pathway (DiLO) vs. without this pathway”.

Ad. 2 We transferred Table 3 and 4 to Supplementary Material.

Ad. 3 Dots were corrected.

Reviewer 2 Report

The authors have done a great job collecting and analyzing data and it is a worthwhile study. The presentation is terrible. The written part is fine but there are simply far too many tables. No reader is going to care about all of the data that went into the study. summarize the data and get rid of most of the tables. At the very least include them in supplemental data. 

Author Response

Thank you very much for reviewing our manuscript and for your kind comments.

Some of tables and figures were transferred to Supplementary material.

Reviewer 3 Report

  • This paper is a report of the activity of two Polish Oncological center during the first year of COVID-19 pandemic. The study's objective was to evaluate how the COVID- 19 pandemic affected the quantity of radiotherapy treatments and the patients who received radical and palliative radiotherapy. 
  • I only suggest to clarify (p18, line 296) what schedules of hypofractionated regimen are used and in which type of tumor

Author Response

Thank you very much for reviewing our manuscript and for your kind comments.

It was 50 Gy in 20 fractions, 2.5 Gy per fraction in selected patients with head and neck cancer not eligible for surgery and standard radical RT and, at the same time, in too good con-dition for standard palliative radiotherapy.

Round 2

Reviewer 2 Report

move all of the numerical tables to supplementary material. they just distract from what you want to show. 

Author Response

All tables were moved to Supplementary material

Round 3

Reviewer 2 Report

thanks for moving the tables